# In silico fragment-based discovery of CIB1-directed anti-tumor agents by FRASE-bot

Yi An[1], Jiwoong Lim[1], Marta Glavatskikh[1], Xiaowen Wang[1,2], Jacqueline Norris-Drouin[1], P. Brian Hardy[1], Tina M. Leisner[1], Kenneth H. Pearce [1] ✉ & Dmitri Kireev [1,2] ✉

Chemical probes are an indispensable tool for translating biological discoveries into new therapies, though are increasingly difficult to identify since novel therapeutic targets are often hard-to-drug proteins. We introduce FRASE-based hit-finding robot (FRASE-bot), to expedite drug discovery for unconventional therapeutic targets. FRASE-bot mines available 3D structures of ligand-protein complexes to create a database of FRAgments in Structural Environments (FRASE). The FRASE database can be screened to identify structural environments similar to those in the target protein and seed the target structure with relevant ligand fragments. A neural network model is used to retain fragments with the highest likelihood of being native binders. The seeded fragments then inform ultra-large-scale virtual screening of commercially available compounds. We apply FRASE-bot to identify ligands for Calcium and Integrin Binding protein 1 (CIB1), a promising drug target implicated in triple negative breast cancer. FRASE-based virtual screening identifies a small-molecule CIB1 ligand (with binding confirmed in a TR-FRET assay) showing specific cell-killing activity in CIB1-dependent cancer cells, but not in CIB1-depletion-insensitive cells.

Recent progress in molecular biology and genome-scale studies constantly increase our understanding of cellular processes and implication of individual proteins in disease[1–5] thus extending the landscape of potential drug targets to harder-to-ligand proteins such as transcription factors, signaling or scaffolding proteins[6–10]. The global drug discovery pipeline is yet to adapt to such a shift. Screening collections used by pharmaceutical companies are mainly composed of ligands targeting historic target families, such as G-protein coupled receptors[11] or protein kinases[12]. Several strategies provide a promising avenue to address the challenge by extending the screenable ligand space. The modern computing prowess enables an ever-increasing scale (up to a billion ligands) of structure-based virtual screening[13–15]. Alternatively, DNA-encoded libraries (DEL) push the boundaries of the accessible chemical space even further by allowing one-pot synthesis and screening of multiple billions of compounds[16–20]. Finally, generative neural networks[21–24] promise access to virtually limitless chemistries. It

is certain, however, that the above strategies come at a cost and bring their own set of problems. For instance, docking and scoring on novel targets are often subject to inacceptable false-positives rates[25,26], multi-billion DEL screening requires a long and costly triage and hit confirmation process, and generative approaches may need a prohibitively time- and effort-consuming synthetic effort without necessarily producing a novel chemistry[22,27,28]. Hence, there is urgent need for both improving the existing strategies and developing new approaches to lead finding.

To further expand the lead finding toolkit, we introduce FRASE-bot, a technology platform for de novo construction of small-molecule ligands to a protein of interest directly in its binding pocket. The only input FRASE-bot needs to initiate the design process is a 3D structure of the protein of interest. It makes use of deep learning to distill 3D information relevant to the protein of interest from 3D structures of tens of thousands of ligand-protein complexes. FRASE-bot exploits the

[1]Center for Integrative Chemical Biology and Drug Discovery, UNC Eshelman School of Pharmacy, University of North Carolina, Chapel Hill, NC 27513, USA. [2]Chemistry department, University of Missouri, Columbia, Columbia, MO 65211, USA. ✉e-mail: khpearce@unc.edu; dmitri.kireev@unc.edu

concept of FRAgments in Structural Environments (FRASE)[29]. FRASEs are structural descriptors blending the chemical and protein structure spaces. A single FRASE includes a chemically sound ligand fragment (e.g., a cycle with adjacent acyclic groups) of a protein-bound ligand and the nearby protein residues (within a distance of 4-5 Å). A comprehensive FRASE database was collected from publicly available sources of experimentally determined 3D structures of protein-ligand complexes[29]. Conceptually, FRASE-based design of a new ligand for a given protein involves two steps: (i) identification of structural environments, stored in the FRASE database, that match those in the protein of interest (this step results in an automated seeding of ligand fragments from the matching FRASEs into the target protein), and (ii) inspecting the seeded ligand fragments and combining them into synthetically tractable compounds. The concept of FRASE has been successfully applied to develop potent kinase-targeting in vivo antitumor agents[29], though it was implemented with one important limitation: it only allowed exploiting information within a given protein family, hence precluding the ligand discovery for the majority of novel targets of interest belonging to understudied families. To overcome that limitation, we developed FRASE-based hit-finding robot (FRASE-bot), a computational platform to exploit the 3D structural and SAR data to identify ligands and their respective binding sites in the protein target. The key components of the platform include a FRASE screening algorithm and a machine learning (ML)-based triage of selected ligand fragments. FRASE-bot might be considered as a step toward a "virtual medicinal chemist" capable of conceiving ligands to novel targets based on an unbiased analysis of structural, chemical and biological data.

FRASE-bot was applied to identify potential agents against Triple Negative Breast Cancer (TNBC) by targeting Calcium- and Integrin-Binding Protein 1 (CIB1)[30-32]. Approximately one million new TNBC cases are diagnosed each year globally and 25% of patients die from this aggressive disease within 5 years of diagnosis[33-35]. Current TNBC treatment options are limited to surgery, radiation and systemic chemotherapy, which often fail due to inherent or acquired resistance. The need for new therapeutic approaches for TNBC is therefore urgent. CIB1 has been found to promote cell survival, growth and proliferation in cancer by regulating at least two prominent growth and oncogenic pathways, PI3K/AKT and MEK/ERK. CIB1, a ubiquitously expressed protein, despite its small size and lack of enzymatic activity, regulates a number of cellular processes including calcium signaling, migration, adhesion, proliferation, and survival[36-39]. The functional versatility is mediated by direct interactions with a broad range of binding partners, such as multiple integrin subtypes, serine/threonine kinases, p21-activated kinase 1 (PAK1), apoptosis signal-regulating kinase 1 (ASK1), and polo-like kinase 3 (PLK3)[30]. The molecular recognition of CIB1 by its binding partners was investigated by NMR, circular dichroism, X-ray crystallography, and sequence analyses. The secondary structure of CIB1 is composed of 10 α-helices, 8 of which form 4 EF-hands. EF-hand is a structural motif found in many proteins including all 4 members of the CIB subfamily (CIB1–CIB4), and closely related homologs, such as calmodulin, calcineurin B (CnB), $K_v$ channel-interacting protein 1 (KChIP1), and Salt Overly Sensitive 3 (SOS3; *Arabadopsis thaliana*). The EF-hand motif is typically involved in the coordination of divalent cations, such as $Ca^{2+}$ [40]. $Ca^{2+}$ seem to play an important role in CIB1 folding; apo-CIB1 is a molten globule, while $Ca^{2+}$-bound CIB1 exhibits a more ordered structure[31,41,42]. Structural changes triggered by Ca2 open a large hydrophobic cleft on the surface of CIB1 that is be responsible for CIB1 binding to multiple integrin subtypes (as shown by a combination of NMR, mutagenesis, and biophysical assays) and putatively other binding partners[43-45]. Most recently, this same cleft was shown to bind (with a nanomolar affinity) artificial helical peptides identified by phage display[46] and their cyclized derivatives[47]. However, neither endogenous nor exogenous non-peptide small-molecule CIB1 binders were identified yet (despite a significant high-

throughput screening effort deployed). Hence, CIB1 is an ideal challenging target for a novel hit finding approach since it does not belong to a well-studied protein family.

In this work, we introduce FRASE-based hit-finding robot (FRASE-bot), to advance ligand identification for challenging therapeutic targets. We employ FRASE-bot to identify ligands for CIB1, a protein implicated in triple negative breast cancer. A small-molecule CIB1 ligand identified by FRASE-bot shows specific cell-killing activity in CIB1-dependent cancer cells, but not in CIB1-depletion-insensitive cells.

# Results

## FRASE-bot

The FRASE-bot workflow includes the following steps: First, the FRASE database is screened to identify FRASEs with structural environments potentially matching those in the target protein. At this step, ligand fragments belonging to hit FRASEs are automatically seeded in the target protein. All potential hit FRASEs (typically thousands) are ranked using a "nativeness" score calculated by a neural network model. The model predicts whether a FRASE's ligand fragment has a native-like pose within the target's protein environment. Next, the top-scoring fragments (tens to hundreds) are exploited to obtain drug-like ligands for the target. Currently, two options exist: (i) build one or more pharmacophore models by analyzing densities of pharmacophoric features on the selected ligand fragments and use them to screen databases of commercially available compounds (this option was used in this study); or (ii) apply a generative neural network to enumerate novel ligand structures matching 2D and 3D constraints set by the poses of the seeded fragments. At the final selection step, the pharmacophore screening hits, or generated compounds may be submitted to various filters, such as computational assessment of the binding free-energy, e.g., by MM-PBSA[48,49] or by a visual inspection of ligand poses. Eventually, selected ligands are purchased or synthesized and tested in relevant bioassays.

## FRASE database screening

Finding whether a target protein has residue arrangements similar to those in the database composed of 184,963 FRASEs (see Methods for the database description) is a combinatorially challenging task. There is a large number of ways in which a set of 5 – 15 residues, could be picked from the target protein and there is a countless number of ways in which an environment picked from the target protein could be aligned with a given environment from the FRASE database. This makes a comprehensive enumeration and alignment of hundreds of environments in a target protein to hundreds of thousands environments stored in the FRASE database computationally prohibitive. To address the combinatorial challenge, all protein environments, both in the database and in the target protein, are encoded into a canonical string notation enabling their ultra-fast comparisons. Due to an oversimplified encoding, the screening step produces a significant number of irrelevant screening hits. However, at the next step of the workflow, all the hits are evaluated by a more thorough scoring method.

In the environment-encoding scheme, for each FRASE, we enumerate triplets of protein residues forming approximately equilateral triangles with sides between 8 Å and 12 Å (Fig. 1). Each residue in a triplet is represented by a many-hot 11-bit string expressing its physical properties—charge, H-bond donor/acceptor, aromaticity, size, and hydrophobicity (e.g., position #1 would be set to 1 if the residue has a positive ionizable group, 0 otherwise). Three positions are reserved for respectively size and hydrophobicity (e.g., all three positions are set to 1 for tryptophane and to 0 for glycine). Bitstrings for individual residues are then concatenated into a single fingerprint for the whole triplet. Six distinct fingerprints are being generated for each triplet to make triplet screening invariant to the order of residues. Typically, 6–15 qualifying triplets are being located in a single FRASE. Although

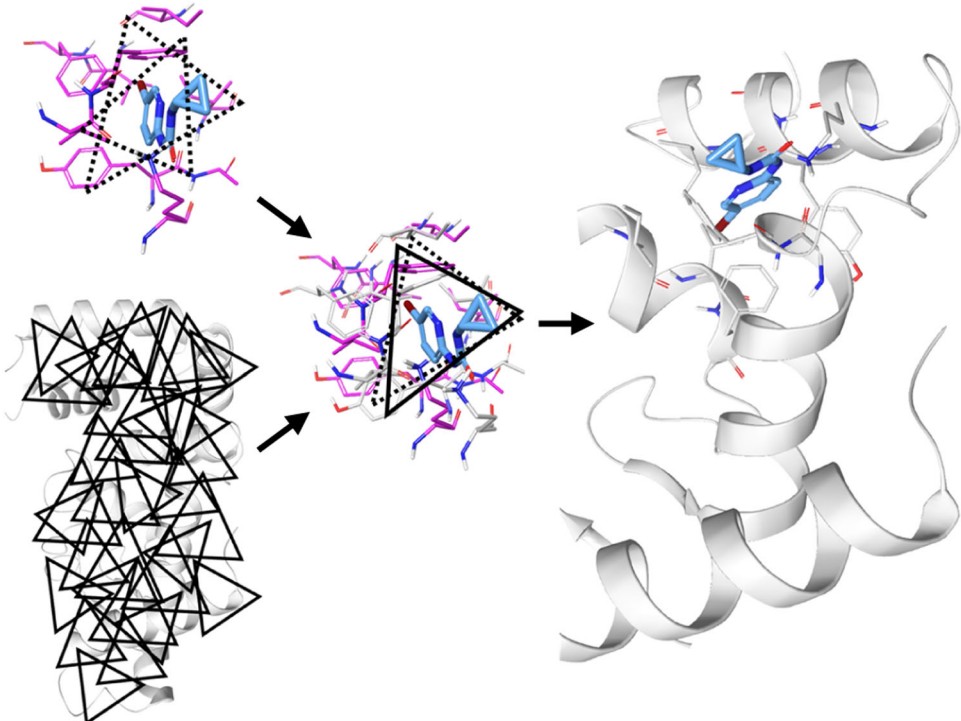

**Fig. 1 | The key algorithm enabling FRASE screening.** Finding whether the structural environment in a FRASE from the database matches a structural environment in the target protein occurs in four steps: (i) each residue triplet in the database's FRASE is compared to each triplet in the target protein; (ii) the matching triplets are used to align the database's FRASE to the protein; (iii) the ligand fragment is used to define the target's FRASE; and (iv) the newly identified target's FRASE is assessed using a machine-learned scoring function on whether it is a proper host for ligand fragment from the database's FRASE.

the triplet fingerprint explicitly encodes only 1D information about the triplets, it also contains an implicit 3D information due to geometric constraints on the shape/size of the triangle formed by the triplet residues. Same procedure can be applied to a whole protein structure, i.e., that of the target protein, to typically produce ~350 triplets for a medium-size protein (233 triplets were enumerated for CIB1). The triplets enable a FRASE database screening process as shown in Fig. 1. First, every triplet of the target protein is compared with every triplet in the database. Second, matching triplets are used to align the respective FRASE with the protein of interest. Third, the ligand fragment from the matching FRASE is inserted into the target protein. Fourth, the ligand fragment is scored on its fitness to the target environment using a neural network model (see next section). One advantage of the FRASE screening approach is that it allows bypassing an explicit detection and comparison of ligand-binding sites, a non-trivial task, and an area of continued research[50–54].

## FRASE scoring

After thousands FRASE hits were selected from the database using fast-search algorithms and ligand fragments seeded in the target protein, each of them is evaluated on their fitness to their new environments. Since it is impossible to experimentally measure contributions of ligand fragments into the potency of the respective ligands, here, the "fitness" is assessed through learning the distinction between interaction patterns in "true" FRASEs (i.e., FRASEs extracted from high-affinity ligand-protein complexes) and interaction patterns in decoy FRASEs. To this end, we applied a neural network model that learns most informative features from the ligand-protein interaction graph to predict the likelihood of whether a FRASE is "true". The interaction graph is an extension of the chemical graph in which every pair of nearby (within 5 Å) ligand/protein atoms is connected by an interaction bond defined by the respective atom-centered pharmacophoric features (see Methods for the feature lists). A full set of interaction

bonds represents a unique ligand-protein interaction signature. To ensure invariance of the interaction signature with respect to the atom numbering, it is transformed into an interaction fingerprint in which any given position encodes a count of interaction bonds of a particular type (e.g., hydrogen-bond donor – hydrogen-bond acceptor). The dimension of the fingerprint is $m*n$ (377), where $m$ (29) is the number of possible pharmacophoric features on the ligand side and $n$ (13) is the number of features on the protein side. The interaction fingerprint is fed to a fully connected feedforward artificial neural network[55] to predict whether an input signature "looks" like a true FRASE or a decoy. The network features two hidden layers (32 and 16 nodes) both employing a Rectified Linear Unit (ReLU) activation function[56] and an output layer activated by a sigmoid function[57], appropriate for binary classification tasks. It was trained using the backpropagation algorithm[57] with a binary cross-entropy loss function[58] and the ADAM optimizer[59], over 500 epochs and a batch size of 50. The training set for this neural network model consists of a set of 38,791 true FRASEs randomly selected from the database and a set of 77,582 decoys. The decoys were produced by randomly swapping ligand fragments between true FRASEs. The trained models were validated by making predictions for the remaining 146,172 true FRASEs and 509,184 decoys.

## Pharmacophore model and screening

To quickly and cost-effectively test the relevance of the fragments seeded in the target protein, we opted for exploiting the fragment-protein complexes in pharmacophore-based virtual screening. To this end, all seeded fragments were converted into pharmacophoric features (Aromatic, Hydrophobic (aliphatic), H-bond Acceptor, H-bond Donor, Positive Ionizable and Negative Ionizable). All features were clustered based on their types and 3D coordinates. Centroids of the most populous and dense clusters were used to compose pharmacophoric queries to search Molport (all in-stock compounds, 2017.2), a database of ~5.1 million commercially available compounds.

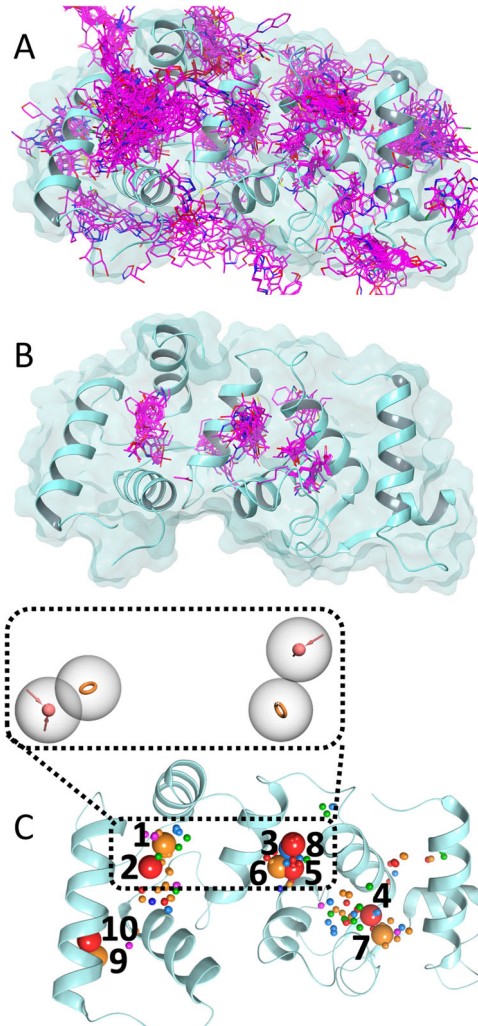

**Fig. 2 | FRASE-based hit generation for CIB1. A** 5362 fragments (rendered in magenta sticks) were seeded into the CIB1 structure (rendered a cyan cartoon) by the initial triplet-based FRASE database screen; (**B**). Application of filters including collision count, buried'ness, and the fitness score further reduced the number of fragments to 151; (**C**). The 151 selected fragments were converted into 398 pharmacophoric features that were clustered into 76 feature clusters (rendered as colored spheres; HBA, red; HBD, light blue; Pos, blue; Ar, orange; Hyd, green). The 10 largest clusters (shown as larger spheres) were considered as potential components of the pharmacophore model. The final model used for screening consisted of 2 HBA and 2 Ar (clusters 1, 2, 6, and 8).

## FRASE database screening and converting hit fragments into pharmacophore queries

Fast screening of the FRASE database was performed against a recent x-ray structure of CIB1 (PDB: 6OCX). This particular structure was chosen because it features a $Ca^{2+}$-bound CIB1 in complex with an exogenous peptide[46] and, hence, the protein is more likely to be pre-formed for ligand binding. The screening enabled identification of 5,362 fragments seeded in the structure's multiple regions (Fig. 2A). Next, a filter passing only those fragments that do not "collide" with the protein (that is, whose atoms are at least 1 Å away from protein atoms) and that are sufficiently "buried" within the protein (that is, each ligand atom has an average of 5 protein atoms within 5 Å) reduced the number of fragments to 726. Finally, the fitness score threshold of 0.4 was applied to further reduce the number of fragments to 151 (Fig. 2B). Each of the 151 hit fragments were then converted into a set of pharmacophoric features (H-bond donor/acceptor (*HBD/HBA*), Ionizable positive/negative (*Pos/Neg*), Aromatic (*Ar*), and Hydrophobic aliphatic

(*Hyd*)), resulting in a total of 398 features (Fig. 2D). All features were clustered using k-means algorithm (see Methods) by type and 3D coordinates into 76 clusters (with a maximum distance between cluster members of 3 Å). Centroids of the ten largest clusters (Fig. 2D) were considered as potential features for the pharmacophore model. Visual inspection of the protein structure suggested that centroids forming two groups (1–2 and 5-8) occupy two distinct nearby pockets that are close enough for a hypothetic ligand combining features from both groups to be a drug-like molecule. Eventually, centroids 1, 2, 6, and 8 were retained to compose a pharmacophore query consisting of two *HBA* and two *Ar* features (Fig. 2D). The centroids 9 and 10 were rejected because they are screened from the remaining clusters by a protein helix and hence no ligands would be able to simultaneously match features on 9/10 and features on any other centroid. We also dismissed centroids 4 and 7 since they are less buried than other centroids. Finally, the features on centroids 1, 2, 6, and 8 were retained to result in a spatially and compositionally balanced pharmacophore query for subsequent virtual screening.

## Virtual screening

The pharmacophoric query was used to screen the MolPort database. Schrodinger's Phase algorithm[60] was used to perform the pharmacophoric search, which yielded 190,320 hits. All pharmacophore hits were docked to the x-ray structure of CIB1 (PDB: 6OCX) using Glide method[61] with standard precision. The docking grid was defined to comprise the four features of the pharmacophore query with a 5 Å margin. The Glide gscore values were distributed in a range from −1.4– −10.3 kcal/mol with a median of 5.6 kcal/mol. About 10,000 ligands (i.e., ~ 5% of the docked set) having gscore below − 7kcal/mol were selected for further triage. These docking hits were clustered to reduce the set of ligands for visual inspection throughout the triage process to top-scored cluster representatives (i.e., marking of a cluster representative for removal meant the removal of the whole cluster). K-means clustering on FCFP4 fingerprints with a maximum Tanimoto distance between cluster members of 0.60 resulted in 567 clusters. The triage process included visual inspection of 2D ligand structures and 3D ligand-protein complexes. First, 567 structures of cluster representatives were scrutinized computationally and visually for the presence of unwanted reactive groups, suitability for docking (e.g., ligands containing long aliphatic linkers were removed), and their lead potential (113 clusters were removed at this step). Second, the 3D poses were browsed for a visual assessment of the binding entropy (e.g., flexible ligands whose binding relies on solvent-exposed hydrogen bonds were removed) and ensuring that the docking pose aligns well with the pharmacophore query (~ 350 clusters were removed at this step). Next, up to 3 cluster representatives (depending on the cluster size) were selected from the clusters that survived the first two triage steps resulting in a set of ~ 200 candidates for purchase. The final triage step involved the refinement of the purchase list based on prices, shipping fees and available budget. Eventually, 56 compounds were purchased for experimental confirmation.

## Experimental confirmation in a CIB1 TR-FRET displacement assay

Binding of the 56 purchased compounds was experimentally assessed in a time-resolved fluorescence energy transfer (TR-FRET) assay (see Fig. 3A and Supplementary data 1). Briefly, it serves as a displacement assay using an AlexaFluor-labeled cyclic CIB1-binding peptide and a Europium (Eu)-labeled Streptavidin donor for attachment to the Avi-Tag biotinylated CIB1 protein (described previously in Puhl et al. and Haberman et al.[47,62]. Dose-response studies (in a 0.005–100 μM concentration range) identified 29 compounds with a dose-dependent response (that is, 52% of virtual hits selected for experimental testing), three of which had $IC_{50}$ values below 10 μM (Fig. 3B). The three most promising compounds were retested in the dose-response TR-FRET

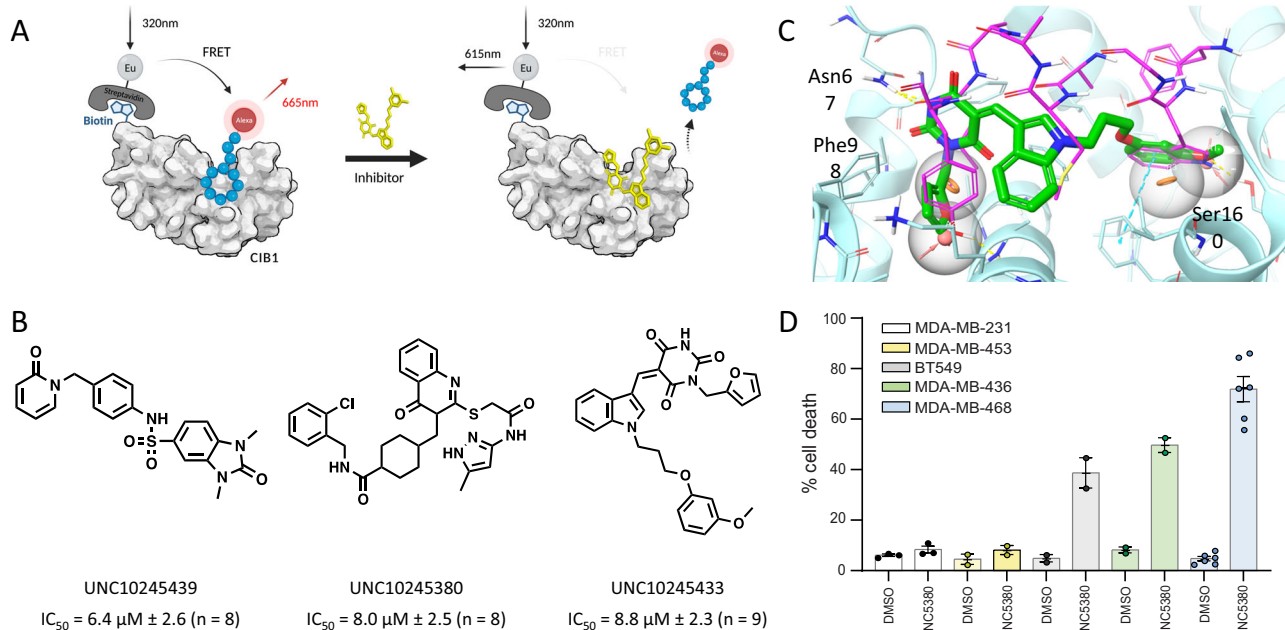

**Fig. 3 | Experimental CIB1 assays. A** CIB1 TR-FRET assay uses an AlexaFluor-labeled cyclic CIB1-binding peptide and a Europium (Eu)-labeled Streptavidin donor for attachment to the AviTag biotinylated CIB1 protein. **B** Three virtual hits showed IC$_{50}$ below 10 μM in CIB1 TR-FRET assays (the IC$_{50}$ values shown are means over the number of replicates ± standard deviation). **C** The x-ray structure of the peptide ligand UNC10245109 (magenta sticks) in complex with CIB1 (cyan sticks and ribbon) (6OCX [https://doi.org/10.2210/pdb6OCX/pdb]), superposed docking pose of UNC10245380 (green sticks) and the pharmacophore query. **D** A panel of CIB1 depletion-sensitive (MDA468, MDA436, BT549) and -insensitive (MDA231 and MDA453) TNBC cells treated with respectively DMSO vehicle or 30 μM UNC10245380 (UNC5380) for 24 h. Cell death was assessed by trypan blue exclusion. UNC5380 selectively kills CIB1 depletion-sensitive TNBC cells. Results are expressed as the mean % of dead cells from adherent and floating cell populations in 3 independent experiments. Error bars correspond to standard errors of the mean (SEM). Source data are provided as a Source Data file.

assay numerous times (n = 8) as biological replicates and the averaged curves are shown in Fig. 3B. Several compounds showed rather high Hill slopes (>3), which is a potential indication of promiscuous or aggregating artifact activity, whereas UNC10245380 typically yielded Hill slopes within the acceptable range of 1.5–2.5. We compared the putative binding pose of UNC10245380 (obtained by Glide docking) with the x-ray structure of CIB1-bound linear peptide UNC10245109 (FWYGAMKALY) (Fig. 3C). UNC10245380 mostly overlaps with the dodecapeptide but occupies a significantly smaller volume. Such an overlap circumstantially supports the hypothetic docking pose, since otherwise the compound would not outcompete the peptide. The respective binding modes feature two similar interactions, hydrogen bonds with respectively Asn67 and Ser160 (though in the interaction with Ser160, the peptide is the hydrogen bond donor, while the small molecule is the hydrogen bond acceptor). On the other hand, UNC10245380 has an unmatched aromatic-aromatic stacking interaction with Phe98. Overall, the common features in binding modes of the peptide and non-peptide ligands suggest that the docking pose is plausible and might be used as a starting point for structure-based lead optimization (until proven wrong).

Additionally, 24 analogs of UNC10245380 were purchased from commercial sources and tested in this TR-FRET assay (see Supplementary data 1). Ten of 24 compounds showed robust dose-dependent response in low-micromolar range (three showed IC$_{50}$ < 10 μM), but none yielded dramatic improvement of potency, and therefore a diverse set of 15 compounds from the initial hit list with strong dose-dependent response in TR-FRET was advanced to cellular assays.

### CIB1 hit selectively target CIB1-dependent cancer cells

Previous studies showed that CIB1 depletion induces cell death and inhibition of CIB1-dependent signaling in 8 of 11 of triple-negative breast cancer cell lines[32]. The remaining three cell lines were insensitive to CIB1 depletion and thus serve as important controls for compound off-target

effects. We selected two CIB1-depletion sensitive and one CIB1-depletion insensitive TNBC cell lines to screen compounds for cell death activity. The three compounds having TR-FRET IC$_{50}$ values below 10 μM were selected for evaluation in TNBC cell lines (see Supplementary data 1). One of the CIB1-sensitive cell lines (MDA-MB-461) was used for primary screening. Two of the selected compounds did not show promising cell killing activity in the MDA-MB-461 cell line and were not advanced to testing in MDA-MB-468 and MDA-MB-231 cell lines. However, one of the tested compounds, UNC10245380, showed significant cell killing activity in both CIB1-depletion-sensitive (MDA-MB-461, and MDA-MB-468) cell lines (Fig. 3D). Importantly, UNC10245380 showed no activity in the CIB1-depletion-insensitive (MDA-MB-231) cell line (Fig. 3D), strongly suggesting a lack of non-specific cellular toxicity. UNC10245380 also showed >90% dose-dependent cell death of MDA-468 TNBC cells over 48 h, indicating a more robust and rapid cell death than CIB1 shRNA depletion by 72 h. We next examined the effects of UNC10245380 on CIB1-dependent signaling events. Western blot analysis showed that UNC10245380 inhibited AKT and ERK phosphorylation in CIB1 depletion-sensitive, but not insensitive cell lines (see Supplementary Fig. 1). UNC10245380 also upregulates death receptor TRAIL-R1/D5 expression specifically in the three CIB1 depletion-sensitive cell lines that recapitulate our previous findings[63] (Supplementary Fig. 1). Taken together, our data indicate that UNC10245380 closely parallels the CIB1 depletion cell death and signaling phenotypes in TNBC cells[32,39,63], and thus provides an appropriate starting point for the future CIB1-targeted anticancer drug development.

### Discussion

In our previous work[29], the concept of fragments in structural environments was introduced and tested in a "model system", that is, on protein targets with multiple known ligands and belonging to a well-established superfamily of protein kinases. Moreover, FRASE-based ligand design was performed stepwise, by visually assessing the choice

of the next fragment and its attachment site. In this study, the FRASE-based ligand finding strategy was applied to Calcium and Integrin Binding protein 1 (CIB1), a target belonging to a small family of four proteins with the closest family member, Calcium and Integrin Binding protein 4 (CIB4), showing only 45% of sequence identity to CIB1. Furthermore, no known non-peptide small-molecule ligands have been reported to any of the CIB family members and their only function appears to be signaling through association with a number of other proteins[43,45,64–66]. Putatively, CIB1 does not interact with any endogenous small-molecule binder and does not have a pocket sculpted by nature for such an interaction. Previously, several unsuccessful attempts have been made by our groups to identify small-molecule CIB1 binders, including high-throughput screening of several collections of ~110 K commercially available compounds[67]. However, the only currently known CIB1 inhibitors are peptides identified from phage display screens[44,46,47]. This study demonstrates the high potential of the FRASE-based strategy in the intended setting, that is, applied to a difficult non-conventional target. We also made use of FRASE-bot in a recent Critical Assessment of Computational Hit-finding Experiments (CACHE) Challenge #1[68,69]. In this competition, the participants were invited to apply their original toolkits to identify hits for LRRK2 WD40 repeat (WDR), a promising Parkinson's target[70–72]. The mechanism by which LRRK2 WDR is implicated in disease, as well as whether and how it can bind a small-molecule ligand, is unknown, thus making it a particularly challenging, high-impact target. Only 7 participating workflows, out of 23, were able to identify any hits. In two rounds of the challenge, we identified 8 experimentally confirmed hits (of 85 submitted compounds, thus showing 9% success rate) with $K_d$ ranging from 3 μM to 44 μM as determined by Surface Plasmon Resonance[73].

On the technology side, FRASE-bot is a formalized semi-automated workflow enabling faster hit finding and leaving less room for a subjective human decision making. The quantum lip from the previously published system[29], in which fragments were only exchanged between the aligned members of the same protein superfamily, has become possible due a triplet-based FRASE screening. Next, the ML-based scoring function enabled fragment ranking by inferring the latent similarity of their 3D poses to those of FRASEs in the FRASE database. Finally, cluster analysis of the fragments' pharmacophoric features made possible a formal algorithm for a transition from a large set of top-scoring ligand fragments to 3D pharmacophore-based screening of millions of commercially available compounds. A possible future alternative to virtual screening might be through using the seeded fragments, along with respective 3D geometric constraints, as an input to ligand generators thus biasing them toward generating binders to the target protein.

Like any empirical method, FRASE-bot has its limitations and caveats. In particular, intrinsic to the approach are hypothetic assumptions throughout the workflow (e.g., that a ligand fragment would necessarily "like" a similar protein environment or that it is possible to learn the difference between true FRASEs and decoys). Because of these assumptions, FRASE-bot can only be used as an enrichment technique (i.e., the generated hit list is expected to show a significantly higher hit rate than a random selection from the same database) rather than a deterministic predictor. Nevertheless, this is a type of enrichment approach that is particularly useful to prescreen billion-scale screening collections, such as Enamine REAL[74], for ligands containing seeded fragments, since substructure search is fast and computationally undemanding. It is also possible that the available protein structure is not in a ligand "friendly" conformation, in which case FRASE-bot would fail to identify hit compounds irrespectively of how efficient the method is.

Conceptually, FRASE-bot builds on the legacy of multiple earlier developments. For over a century, medicinal chemists dealt with finding optimal substituents for a given scaffold, a paradigm quantified through an additive approach by Free and Wilson in 1970 s[75,76]. In 1990 s, the advent of high-throughput x-ray and NMR technologies enabled structure-guided fragment-based discovery that explicitly exploited the concept of fragments "liking" specific protein environments[77–80]. In 2000 s, a broad family of approaches collectively termed chemogenomics or proteochemometrics experimented with various ways of using the protein sequence or structure to share SAR information between targets[81–85]. Finally, during the last decade, deep machine learning has shown potential to combine protein and ligand data for a fast prediction of protein-ligand binding affinity[86–89].

Beyond FRASE-bot, its components may have a broader use in hit finding and drug design. For instance, the ligand fragments identified by FRASE screening can be used as an input to a conventional structure-based design through an incremental fragment growing or plugged as a structural constraint to a generative neural network. Furthermore, the module translating a large number of seeded ligand fragments into pharmacophoric queries for a large-scale screening of commercially available compounds can be applied to experimentally identified fragments, e.g., from the Diamond Light Source[90]. And the neural scoring function distinguishing between true FRASEs and decoys could be used to rank poses in structure-based virtual screening. Another important outcome of this study is a small-molecule CIB1 ligand with a demonstrated cellular effect. We expect it be a helpful probe to further exploit and validate CIB1 as a promising anti-cancer target. Later, UNC10245380 may be a source of inspiration for developing a CIB1-targeting drug or an in vivo probe.

## Methods
### FRASE database
The FRASE database was collected using data-processing protocols implemented in Pipeline Pilot[91] X-ray structures of high-affinity ligand-protein complexes were imported from the Protein Data Bank (PDB)[92]. The PDB codes for high affinity complexes for drug-like ligands were obtained from BindingDB[93] (retrieved in 05/2018). "High affinity" was defined as $K_D$, $K_i$, $IC_{50}$ or $EC_{50} < 100$ nM. Drug-likeness of the ligands was warranted by filters including Lipinski[94], REOS[95] and structural queries to remove peptides, inorganic and phosphorous compounds, as well as those containing highly reactive groups. This selection process resulted in 10,464 complexes involving 4724 unique ligands and 3,068 unique proteins. This dataset allowed us to generate a database 184,963 FRASES involving 51,060 unique ligand fragments. A FRASE was defined as a ligand fragment with all nearby protein residues (i.e., residues having at least one atom within 4.5 Å from the closest ligand's atom). Ligands were fragmented using the "Enumerate Fragments" Pipeline Pilot component, allowing only single, non-cyclic bonds to be broken and keeping α-atoms attached to the cyclic fragments. Only fragments weighing between 50 and 300 Da were retained. All FRASEs, that is, ligand fragments with nearby protein residues, were saved to an SD file.

### FRASE database screening
The screening protocols used in this study were implemented in Pipeline Pilot[91] as a combination of standard components and custom Pilot scripts. The four key steps of the screening process include (i) enumerating triplets of protein residues and representing them as fingerprints, (ii) similarity search in the FRASE database for triplets similar to those in the target protein, (iii) alignment of the FRASEs containing hit triplets to the protein structure by the $C_\alpha$ atoms of the triplet's residues, and (iv) making a target-based FRASE for further scoring. The "Align Molecules using Substructure" component is used for the alignment. After the hit FRASE from the database is aligned to the target protein, its ligand fragment can be used to create a new, target-based FRASE by cutting out the nearby residues of the target protein (i.e., residues having at least one atom within 4.5 Å from the closest ligand's atom) and merging them with the ligand fragment.

All triplets of protein residues satisfying the geometric condition were enumerated in all FRASEs from the database and in the target protein. The geometric condition for a set of three residue to qualify as

**Table 1 | Bit strings for 20 amino acids used in residue triplet screening**

| Residue name | Code | Positive ionizable | Negative ionizable | H-bond acceptor | H-bond donor | Aromatic | Hydrophobic | Volume |
|---|---|---|---|---|---|---|---|---|
| Alanine | ALA | 0 | 0 | 0 | 0 | 0 | 1 0 0 | 1 0 0 |
| Arginine | ARG | 0 | 1 | 0 | 0 | 0 | 0 0 0 | 1 1 1 |
| Asparagine | ASN | 0 | 0 | 1 | 1 | 0 | 0 0 0 | 1 1 0 |
| Aspartic Acid | ASP | 1 | 0 | 0 | 0 | 0 | 0 0 0 | 1 1 0 |
| Cysteine | CYS | 0 | 0 | 1 | 1 | 0 | 1 1 0 | 1 1 0 |
| Glutamic Acid | GLU | 1 | 0 | 0 | 0 | 0 | 0 0 0 | 1 1 1 |
| Glutamine | GLN | 0 | 0 | 1 | 1 | 0 | 0 0 0 | 1 1 0 |
| Glycine | GLY | 0 | 0 | 0 | 0 | 0 | 0 0 0 | 0 0 0 |
| Histidine | HIS | 0 | 1 | 0 | 1 | 1 | 0 0 0 | 1 1 0 |
| Isoleucine | ILE | 0 | 0 | 0 | 0 | 0 | 1 1 0 | 1 1 0 |
| Leucine | LEU | 0 | 0 | 0 | 0 | 0 | 1 1 0 | 1 1 0 |
| Lysine | LYS | 0 | 1 | 0 | 0 | 0 | 0 0 0 | 1 1 1 |
| Methionine | MET | 0 | 0 | 0 | 0 | 0 | 1 1 1 | 1 1 1 |
| Phenylalanine | PHE | 0 | 0 | 0 | 0 | 1 | 1 1 1 | 1 1 1 |
| Proline | PRO | 0 | 0 | 0 | 0 | 0 | 1 0 0 | 1 0 0 |
| Serine | SER | 0 | 0 | 1 | 1 | 0 | 0 0 0 | 1 1 0 |
| Threonine | THR | 0 | 0 | 1 | 1 | 0 | 1 1 0 | 1 1 0 |
| Tryptophan | TRP | 0 | 0 | 0 | 1 | 1 | 1 1 0 | 1 1 1 |
| Tyrosine | TYR | 0 | 0 | 1 | 1 | 0 | 1 1 0 | 1 1 1 |
| Valine | VAL | 0 | 0 | 0 | 0 | 0 | 1 1 0 | 1 1 0 |

a triplet was for its respective C$_\alpha$ atoms to form an approximately equilateral triangle with edge lengths between 8 Å and 12 Å. Each residue in a triplet is represented by a many-hot 11-bit string expressing its physical properties—charge, H-bond donor/acceptor, aromaticity, size, and hydrophobicity. First 3 positions are reserved for size, next 3 for hydrophobicity, 1 for hydrogen bond acceptors, 1 for hydrogen bond donors, 1 for positive ionizable, 1 for negative ionizable, and 1 for aromatic. Bit strings for all 20 side chains are shown in Table 1.

### FRASE scoring

FRASE scores were calculated using a neural network model for binary classification. The model is trained to learn the difference between true FRASEs and decoys. The decoys were generated by randomly shuffling ligand fragments between true FRASEs. FRASEs and decoys were represented by fixed-length feature vectors. Each feature corresponds to a distance-weighted pair of atom types centered, respectively, on a ligand and a protein atoms. The 29 ligand atom types are "Aromatic", "Hetero", "Halogen", "Negative Ionizable", "Positive Ionizable", "H-bond Donor", "H-bond Acceptor", "Amide Nitrogen", "Amine Nitrogen", "Vinyl", "Carboxylate Oxygen", "Alcohol Oxygen", "Nitro Oxygen", "Nitro Nitrogen", "Phosphate Oxygen", "Sulfone Sulfur", "Sulfoxide Sulfur", "Enol Oxygen", "Imine Nitrogen", "Enamine Nitrogen", "Aromatic Nitrogen", "Aromatic Oxygen", "Aromatic Sulfur", "Aromatic Carbon", "Aliphatic Carbon", "F", "Cl", "Br", and "I". The 13 protein atom types are "Aromatic", "NegativeIonizable", "Positive Ionizable", "H-bond Donor", "H-bond Acceptor", "Amide Nitrogen", "Amine Nitrogen", "Carboxylate Oxygen", "Alcohol Oxygen", "Imine Nitrogen", "Aromatic Nitrogen", "Aromatic Carbon", and "Aliphatic Carbon". Hence, the dimension of this interaction feature vector is 377. The value on a feature is calculated as follows:

$$f_{mn}^{l} = \sum_{i=1}^{N_{atm}^{lig}} \sum_{j=1}^{N_{atm}^{prot}} \delta_{mi} \delta_{nj} w(d_{ij}) \qquad (1)$$

where $f_{mn}^{l}$ is the $l$-th element of the feature vector corresponding to the pair of the ligand atom type $m$ and the protein atom type $n$, $\delta_{mi}$ is the Kronecker delta equal to 1 if the $i$-th atom is of type $m$ (otherwise,

0), and $w(d_{ij})$ is the distance weighting factor equal to

$$\begin{cases} e^{-(d_{ij}-3)^2}, \text{if } d_{ij} \le 10 \\ 0, \text{if } d_{ij} > 10 \end{cases} \qquad (2)$$

where $d_{ij}$ is distance between the ligand atom $i$ and the protein atom $j$.

### Pharmacophore queries and screening

The 3D pharmacophore queries for database screening were created using the Phase software[96] with Maestro graphics interface. Individual pharmacophore features were created and manually placed at the centers of feature clusters (see "Pharmacophore model and screening" in the main text for context). The created features were merged into pharmacophore queries for screening in the "Merged hypotheses" mode. The ligand input file was in maegz format. Prior to screening, the Molport ligand collection (all in-stock compounds, 2017.2) was filtered through a modified Lipinski[94] and REOS[95] filters (the modified Lipinski rule allowed ligands of up 600 Da). The 3D ligand structures for 5,142,498 ligands from the MolPort database were generated by the Pipeline Pilot software[91]. The "Generate conformers during search" (up to 50) was applied to the input ligands. "PhaseScreenScore" was used to rank the screening hits.

### Docking

Ligands were docked to CIB1 using the Glide program[61] in standard docking precision (Glide SP). The binding region was defined by a 20 Å × 20 Å × 20 Å box centered on the geometric center of the pharmacophore model. A scaling factor of 0.8 was applied to the van der Waals radii. Default settings were used for all the remaining parameters. One pose per ligand was generated.

### Virtual screening triage

Virtual hits ranked and selected based on the Phase Fitness score and Glide gscore were submitted to a hit triage process. First, the hit redundancy was reduced through retaining the best-scoring hits from clusters of similar compounds. Clustering was performed by a k-means method as implemented in the Pipeline Pilot software[91]. The inclusion criterion was 45% of Tanimoto similarity on ECFP4 fingerprints to the current

cluster center. Second, binding poses of top-ranked non-redundant hits were visually inspected to remove poses whose scores putatively underestimate the entropic penalty on binding. Finally, a fraction of hits from the final list was eliminated based on pricing and availability criteria.

### TR-FRET assay
For the TR-FRET tracer molecule, UNC10245204 (a cyclic peptide -S-Ac-YTTPIWNIRFC-NH2 described in ref. 47) was synthesized with an N-terminal cysteine to facilitate conjugation with maleimide Alexa-Fluor 647 (Thermo Fisher Scientific; Alexa647-CIB1-peptide). Compounds were dispensed into 384-well plates using a Mosquito HTS nanoliter instrument (TTP LABTech) as 3-fold serial dilutions (100x in DMSO, 0.1 μL) for final concentrations ranging from 100 μM to 0.5 nM. Biotinylated CIB1 was diluted to a final concentration of 3 nM in assay buffer (20 mM) TRIS pH 7.5, 150 mM NaCl, 1 mM CaCl, 1 mM CHAPS, and 1 mM DTT (added fresh each time), and 5 μL of diluted protein was added to the wells of the assay plate using a Multidrop Combi Reagent Dispenser (Thermo Scientific) and incubated at room temperature for 20 min. 5 μL detection solution containing 2 nM Lance Eu-Streptavidin (Perkin Elmer) and 30 nM Alexa647-CIB1-phage peptide diluted in assay buffer was added to the wells and incubated 30–60 min at room temperature protected from light. TR-FRET signals were measured using an Envision Multilabel Plate Reader (PerkinElmer; Eu excitation 320 nm, Eu emission 615 nm, Alexa dye emission 665 nm). TR-FRET signal is measured as the ratio 665 nm/615 nm, and percent inhibition was calculated using two sets of control wells; biotinylated CIB1 and detection solution in the presence (100% inhibition, positive control) or absence (0% inhibition, negative control) of CIB1 inhibitor. Inhibition curves were analyzed using a four-parameter non-linear curve fit using ScreenAble software, and the mean and standard deviation were calculated. Publication curves were averaged and fit with a four-parameter non-liner curve fit using GraphPad Prism.

### Cellular assay
Human triple-negative cell lines MDA-MB-468, MDA-MB-436, MDA-MB-231, and BT-549 were cultured in Dulbecco's modified eagle medium (DMEM, Gibco) supplemented with 10% fetal bovine serum and 1% non-essential amino acids (Gibco) at 5% $CO_2$ and 37 °C. BT549 and MDA-436 media was also supplemented with 10 μg/ml insulin (Gibco). For cell death studies, each cell line was plated at a density of $1.5 \times 10^5$ cells/well. After 24 h, the media was replaced with 1.5 mL of media containing 30 μM of compound or 1% DMSO vehicle and incubated and additional 24–48 h. Floating and adherent cell populations were harvested and cell death quantified by Trypan blue exclusion and expressed as the mean percentage of dead cells (i.e., trypan blue positive) from both floating and adherent total cell populations. Statistical analysis from 2 separate experiments was performed using GraphPad Prism software.

### Reporting summary
Further information on research design is available in the Nature Portfolio Reporting Summary linked to this article.

## Data availability
The input/output files generated by the Pipeline Pilot workflow are shared as a Mendeley data set (https://doi.org/10.17632/9yn47cy5jv.1). A spreadsheet with TR-FRET data for 56 compounds selected by virtual screening, as well as for 24 compounds resulting from the SAR-by-catalog study is shared as a Supplementary data 1. Source data for the figures are provided with this paper. Source data are provided with this paper.

## Code availability
The source code used to perform the current study is shared as a Supplementary archive file and through the GitHub repository https://github.com/kireevlab/FRASE-bot-Pipeline-Pilot[97]. The latest version of FRASE-bot implemented as Python code is available at https://github.com/kireevlab/FRASE-bot-RDKit[98].

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

## Acknowledgements

We thank all the members within the Center for Integrative Chemical Biology and Drug Discovery for advice and technical support. This work was supported by grant 5R01GM132299 from NIH, National Institute of General Medical Sciences (NIGMS) to D.K. We acknowledge the University of Missouri, Division of IT, Research Computing Support Services for the use of the computing resources.

## Author contributions

Methodology, Investigation, Analyses, Writing, & Editing: Y.A., M.G., J.L., X.W., J.N-D, B.H., T.M.L., K.H.P., and D.K.; Conceptualization, Writing, Review, Supervision, Resources, Project Administration: K.H.P. and D.K.

## Competing interests

The authors declare no competing interests.
