## [Peer Review File · Nature Communications]

In silico Fragment-based Discovery of CIB1-directed
Anti-Tumor Agents by FRASE-botREVIEWER COMMENTS

Reviewer #1 (Remarks to the Author):

In the manuscript by An et al. low-millimolar hits against CIB1 in a starting round (without any prior empirical hits) are identified using their algorithm.

FRASE-bot is a virtual fragment generating algorithm, which generated 151 pharmacophoric hotspots, which were used for virtual screening with Schrödinger Phase and Glide docking. This gave an impressive 19 compounds out of 56 producing a signal in their assay.

However, three essential points need addressing: the titular neural network needs to be a lot better explained (especially Figure 1), the data needs to be released and lastly the biochemistry of the target should not be obfuscated.

Neural network

> FRASE-bot exploits big data and machine learning (ML) to distill 3D information relevant to the target protein from thousands of protein-ligand complexes to seed it with ligand fragments. Based on homophony with Phrase, I incorrectly guessed that FRASE-bot was an LLM. Please explain in the abstract what you mean by FRASE. Furthermore, that quoted statement is extremely vague.

Uncommonly for a manuscript demonstrating a method with empirical validation, the wet lab experiments do stifle the algorithmic focus. However, the neural network is extremely vague. <https://github.com/kireevlab/FRASE-bot-RDKit> is an empty repository. And the model is simply referred to as a binary-classifier "neural network" trained on true and decoy values. Upon initial reading, I could not determine if it were a Multilayer Perceptron (MLP), another type of Feedforward Neural Network (FNN), a Convolutional Neural Network (CNN) or something else. Buried within the methods I understood it was probably a MLP with two hidden layers (32 and 16 nodes) with a rectified linear activation function (ReLU) and an output layer with a sigmoid activation function and a binary cross-entropy loss function. To train the network, the backpropagation process was used and emphasised in the manuscript. Please, be a lot more explicit about your model. Figure 1B needed to be redesigned.

Regarding the empty repository, is this work open, and which license will be used?

Missing biochemical context

I understand that the target, CIB1, is a methodological example and it is great that well-trodden target was avoided purposefully, but I was honestly frustrated by the fact that protein, CIB1, is being sold as a complete mystery and all in silico biochemistry is avoided to push this angle.

In the manuscript CIB1 is described at the cell biology level (i.e. target identification), but not at the biochemical level: the former treats activation as a magical arrow, whereas in the latter is a molecular-mechanics-driven by conformational changes, which is important for drug discovery. The template used is a PDB structure (6ocx) deposited by a subset of the authors of this manuscript, from a paper where phage display is used to find a peptide inhibitor. Therein, the biochemistry of the target are described.

I am not familiar with CIB1, but calcium and integrin are in its name. Calcium is physically bound in the template. This would represent the calcium-bound state. Was this state methodologically preferable to the apo-state?

From a quick glance there is no human CIB1 structure bound to integrin. I performed a BLASTP in NCBI of NP_001264693.1 against the 'PDB' sequence dataset and got pages of hits. The orthologue Calmyrin in *C. elegans* (40% id) in 7USW is in a complex with integrin, which seems to binds where

tagged-decapeptides, UNC10245092 and UNC10245109 bind. Were this unavailable (released last year), AlphaFold2 or Colabfold multimer might have been used to glean where the sites of interest are. I mention this because inhibitors are not compounds that bind anywhere in a protein, but in a specific location (the PPI with integrin in this case). In the manuscript all sites are brought forth (cf. figure 2B and C) and yet for the detection to work on the inset of figure 2C (or 3D see minor) was relevant, which is fine because integrin likely binds there anyway.

On this front the following statement is perplexing:

> Putatively, CIB1 does not interact with any endogenous small-molecule binder and does not have a pocket sculpted by nature for such an interaction.

This is an EF-hand protein that binds calcium, which is an ion not a small-molecule, but is nevertheless a well recognised cytosolic signal in Eukaryotes.

A big issue with some neural networks is that they often "remember" their training data and simply regurgitate variants. As the neural network presented here is used to generate pharmacophoric hotspots and not candidate compounds, this concern does not apply. Although, I would be interesting in knowing if possible how many FRASE pharmacophoric triangles are shared with a subset of other `_structural_` homologues.

Assay

Could any processed binding curves be shown, especially those with high Hill slopes and the three leads.

Are any of the fluorescent or quenching? And what controls were run?

Docking

Once the fragment hotspots were identified, Schrödinger Phase and Glide were used to enumerate virtual hits. This raises the question of how much did FRASE-bot contribute to this. Requesting that a control be done is an onerous request. Furthermore, 3 out of 56 is $5\pm 3\%$, which is decent for docking —19/56 is $33\pm 5\%$, which is better than even elaborations of first round hits, albeit with the caveat that not all may be true positives for displacement. So I sense the answer is that FRASE did contribute. However, if my fellow reviewers have raised similar concerns, please consider this as an underlining of their concerns.

Minor points:

* Throughout the manuscript "FRASE" is used as a given concept, please correct this for non-linear readers, especially for those who jump into the figure legends

* Please elaborate in the results on what UNC10245204 is, eg. lab-evolved decapeptide CIB1 binder like UNC10245092 and UNC10245109 or fragment of integrin etc. Also if it's linear why is the blue Alexa-labelled pearl-string cyclical in the figure.

* The text mistakenly refers to Figures 2B and 2C as Figure 3. Additionally, the inset in Figure 2C is incorrectly labeled as '3D'.

* Please specify that the sequence identity to its closest paralogue is `_AA_`, it is obvious and correct, but avoids any doubt in sceptical readers.

Reviewer #2 (Remarks to the Author):

The manuscript "Machine Learning-driven Fragment-based Discovery of CIB1-directed Anti-Tumor Agents by FRASE-bot" describes the identification in silico and experimental validation of the first

small molecule inhibitor targeting CIB1, an important protein for the treatment of cancer. The manuscript is complete, well written, and the results described have a significant importance in the field.

The FRASE-bot protocol described in the manuscript is presented as generally applicable for orphan targets (in contrast with the kinases, where it's relatively easy to de-orphanize a new kinase with structural data of ligands from other kinase families). However, I have some objections regarding the general applicability of this protocol.

1. CIB1 is not truly an orphan target, as there are several known peptides (identified by the same group) targeting this protein. I suspect that the authors have used in the selection process described in this paper some structural information derived from the complexes of these peptides with CIB1. The authors should include in the manuscript a structural comparison of the small molecules inhibitors from this study with the previously identified peptides, with the pharmacophoric features selected during the workflow highlighted.
2. Related to 1., a TR-FRET assay based on these peptides was used for the experimental validation of the initial hits, which would not be possible for a real orphan target.
3. The FRASE-bot procedure is very complex, involving tools from two different modeling packages (PipelinePilot from BIOVIA and Phase and Glide from Schrödinger). Why it was not possible to use KNIME from Schrödinger instead of PipelinePilot ? Additionally, there are many steps involving human intervention and subjective decisions in the selection process, which makes this protocol difficult to apply to other targets.
4. In my opinion, the straightest approach to identify new small molecule inhibitors of CIB1 is to design peptidomimetics, based on the structure of the previously identified peptides. Did the authors attempted this approach ? If yes, what was the outcome ? Also, it would be useful for the community to specify the names of "collections of ~110K commercially available compounds" (page 8), to avoid their useless evaluation by other groups.
5. Machine learning is indeed involved in the first step of the selection process, but its role doesn't justify to use it in the title of the manuscript (docking and pharmacophoric search are also essential for the successful outcome of the selection process).

Minor comments:

p3: "Several 3D structures of apo and ligand bound CIB1" - specify that these are peptidic ligands, not small molecules

p4, caption Figure 1A: replace "(vi)" by "(iv)"

Supplementary Excel file:

- replace "Smiles" by "SMILES"

- most SMILES strings are incorrect, please check and replace them with the correct version. For example, for the first structure it should be

COC1=CC(=CC=C1)OCCCN2C=C(C3=CC=CC=C32)/C=C\4/C(=O)NC(=O)N(C4=O)CC5=CC=CO5.

Also, use Isomeric SMILES instead of Canonical SMILES to avoid losing important stereochemical information.

Reviewer #3 (Remarks to the Author):

In this manuscript, the authors report on their study on using FRASE-based ligand binding strategy to find ligands for the Calcium and Integrin Binding Protein 1 (CIBP1), an orphan target with no known ligand.

Using this strategy, the authors identified UNC10245380 as the most promising compound. The authors tested the target hit selectivity of the compound in a set of triple negative breast cancer (TNBC) cells whose CIBP1-dependency for growth were reported by another laboratory. Based on a preliminary cell death data from trypan blue exclusion assay (shown Fig. 3C), and western blot of AKT and ERK phosphorylation (the authors did not show the western blot data), the authors claim to have

showed specific cell-killing activity in CIB1-dependent TNBC cells. To substantiate the claim, the authors should perform orthogonal assays for cell death (for example, PARP-cleavage, Annexin V staining) that can be performed with these cell lines.

The sentence in the Abstract ".....showing specific cell-killing activity in CIB1-dependent cancer cells, but not in CIB1-depleted cells" is scientifically not precise. It should be "..... CIB1-depletion-insensitive cells" as was used on page 7 of the manuscript.

In the paper the current manuscript has referred to (#32, Black et al., Breast Cancer Res Treat, 152:337-346(1915), the authors there did not say that the TNBC cells that are insensitive are depleted of CIB1 (see western blot in Fig. 1B in that paper where insensitive cells do express CIB1 similar to the sensitive cells). Therefore, it is important to phrase the sentence appropriately.

Minor comment: There is no Fig. 3 in the manuscript. Incorrectly, the Fig is labeled Figure 4.

We thank the reviewers for their feedback that helped us to significantly improve the manuscript. Below is a point-by-point response to the reviewers' concerns (the original reviewers' text is rendered in blue font). In the manuscript file, the new or modified text is highlighted in yellow.

Reviewer #1:

However, three essential points need addressing: (1) the titular neural network needs to be a lot better explained (especially Figure 1), (2) the data needs to be released and (3) lastly the biochemistry of the target should not be obfuscated.

The concerns #1 and #3 are addressed below, in response to more specific remarks. Regarding #2, we rewrote the SUPPLEMENTAL INFORMATION section making it more clear which data/source code are shared. In particular, the repository <https://github.com/kireevlab/FRASE-bot-Pipeline-Pilot> (this link was broken in the manuscript pdf) contains all the Pipeline Protocols used in this study as well as the neural network model. This repository has a readme section explaining how to run the protocols and the model. All the input/output files produced by the protocols are shared as a Mendeley data set <http://dx.doi.org/10.17632/9yn47cy5jv.1>. The combination of the source code and the data files allows on to reproduce every step of the workflow described in the manuscript. The repository <https://github.com/kireevlab/FRASE-bot-RDKit> is not empty anymore. It contains the latest version of FRASE-bot in the form of Python scripts that do not need any commercial licenses to be run. However, this repository is constantly being updated and, hence, the results generated by the Python version would not be an exact replica of the manuscript results.

“FRASE-bot exploits big data and machine learning (ML) to distill 3D information relevant to the target protein from thousands of protein-ligand complexes to seed it with ligand fragments.” Based on homophony with Phrase, I incorrectly guessed that FRASE-bot was an LLM. Please explain in the abstract what you mean by FRASE. Furthermore, that quoted statement is extremely vague.

A more specific FRASE-bot description in the abstract now reads: “We introduce a computational platform, FRASE-bot, to expedite drug discovery for unconventional therapeutic targets. FRASE-bot mines available 3D structures of ligand-protein complexes to create a database of FRAGments in Structural Environments (FRASE). The FRASE database can be screened to identify structural environments similar to those in the target protein and seed the target structure with relevant ligand fragments. A neural network model is used to retain fragments with the highest likelihood of being native binders. The seeded fragments can”.

The neural network is extremely vague. <https://github.com/kireevlab/FRASE-bot-RDKit> is an empty repository. And the model is simply referred to as a binary-classifier “neural network” trained on true and decoy values. Upon initial reading, I could not determine if it were a Multilayer Perceptron (MLP), another type of Feedforward Neural Network (FNN), a Convolutional Neural Network (CNN) or something else. Buried with in the methods I understood it was probably a MLP with two hidden layers (32 and 16 nodes) with a rectified linear activation function(ReLU) and an output layer with a sigmoid activation function and a binary cross-entropy loss function. To train the network, the backpropagation process was used and emphasised in the manuscript. Please, be a lot more explicit about your model. Figure 1B needed to be redesigned.

We extended the NN description, supplied it with references, and moved to the Results section (see below how it reads). We believe that the new version is complete enough for the model to be reproduced. Moreover, the Python code of the model is shared through the repository <https://github.com/kireevlab/FRASE-bot-Pipeline-Pilot>.

“The interaction fingerprint is fed to a fully connected feedforward artificial neural network⁴⁵ to predict whether an input signature “looks” like a true FRASE or a decoy. The network features two hidden layers (32 and 16 nodes) both employing a Rectified Linear Unit (ReLU) activation function⁴⁶ and an output layer activated by a sigmoid function⁴⁷, appropriate for binary classification tasks. It was trained using the backpropagation algorithm⁴⁷ with a binary cross-entropy loss function⁴⁸ and the ADAM optimizer⁴⁹, over 500 epochs and a batch size of 50.”

Regarding the empty repository, is this work open, and which license will be used?

FRASE-bot is licensed under the MIT License.

I understand that the target, CIB1, is a methodological example and it is great that well-trodden target was avoided purposefully, but I was honestly frustrated by the fact that protein, CIB1, is being sold as a complete mystery and all in silico biochemistry is avoided to push this angle.

In the manuscript CIB1 is described at the cell biology level (i.e. target identification), but not at the biochemical level: the former treats activation as a magical arrow, whereas in the latter is a molecular-mechanics–driven by conformational changes, which is important for drug discovery.

The CIB1 description (in the Introduction section) was significantly extended and now includes an overview of the protein's structural and biophysical characterization.

The template used is a PDB structure (6ocx) deposited by a subset of the authors of this manuscript, from a paper where phage display is used to find a peptide inhibitor. Therein, the biochemistry of the target are described. I am not familiar with CIB1, but calcium and integrin are in its name. Calcium is physically bound in the template. This would represent the calcium-bound state. Was this state methodologically preferable to the apo-state? From a quick glance there is no human CIB1 structure bound to integrin. I performed a BLASTP in NCBI of NP_001264693.1 against the 'PDB' sequence dataset and got pages of hits. The orthologue Calmyrin in *C. elegans* (40% id) in 7USW is in a complex with integrin, which seems to binds where tagged-decapeptides, UNC10245092 and UNC10245109 bind. Were this unavailable (released last year), AlphaFold2 or Colabfold multimer might have been used to glean where the sites of interest are. I mention this because inhibitors are not compounds that bind anywhere in a protein, but in a specific location (the PPI with integrin in this case). In the manuscript all sites are brought forth (cf. figure 2B and C) and yet for the detection to work on the inset of figure 2C (or 3D see minor) was relevant, which is fine because integrin likely binds there anyway. On this front the following statement is perplexing: "Putatively, CIB1 does not interact with any endogenous small-molecule binder and does not have a pocket sculpted by nature for such an interaction.". This is an EF-hand protein that binds calcium, which is an ion not a small-molecule, but is nevertheless a well-recognized cytosolic signal in Eukaryotes.

In the new version of CIB1 introduction, we highlight the role of Ca^{2+} ions in shaping the CIB1 structure. The evidence (a combination of x-ray, NMR, mutagenesis, and biophysical data, now cited in the manuscript) suggests that Ca^{2+} binding induces a large hydrophobic cleft on the CIB1 surface which binds the integrin αIIb cytoplasmic tail (and putatively other binding partners). Hence, 6OCX, which is both Ca^{2+} - and peptide-bound structure, is a likely small-molecule ligand binder too. This probably not the only ligand-binding conformation and a thorough molecular dynamics study could have revealed more, but such an analysis would be beyond the scope of this manuscript. We added a rationale for the choice of 6OCX in the beginning of the Results section. Also, we replaced the quoted sentence with a less speculative one "However, neither endogenous nor exogenous non-peptide small-molecule CIB1 binders were identified yet".

A big issue with some neural networks is that they often "remember" their training data and simply regurgitate variants. As the neural network presented here is used to generate pharmacophoric hotspots and not candidate compounds, this concern does not apply. Although, I would be interesting in knowing if possible how many FRASE pharmacophoric triangles are shared with a subset of other structural homologues.

As noted in the manuscript, we consider the triplets only as intermediary data, insignificant on their own, serving mainly to align a FRASE's structural environment with that of the target protein. Typically, only ~3% matching triplets survive after all downstream filters (including the neural network model) are applied. Consequently, we did not develop any tools for an in-depth analysis of triplet-based properties, such as a triplet-based similarity between proteins. However, a reasonably accurate estimate of shared triplets is possible. In particular, we know that there is 45% sequence identity between CIB1 and its closest homolog CIB4 and based on a visual assessment, the matching residues are evenly spread throughout the sequence. Then, enumeration of triplets on a random subset of 45% of CIB1 residues gives rise to an average of 26 triplets. The latter figure is a sound proxy for the number of shared CIB1/CIB3 triplets, that is, 11% of the total number of (233) CIB1 triplets. In case, we would consider residue similarity rather than identity and set the similarity threshold to a value that would result in 70% of identical or similar CIB1/CIB3 residues, then the average number of shared triplets is 100, that is a bit less than a half of all CIB1 triplets.

Could any processed binding curves be shown, especially those with high Hill slopes and the three leads. Are any of the fluorescent or quenching? And what controls were run?

The binding curves for the three leads are now included in Fig. 3. We believe that including the curves with high Hill slopes would not be of high interest for the general readership since those compounds were anyway eliminated from further consideration since we suspect them to be fluorescence or aggregating artifacts. We did not directly measure

fluorescence or quenching by the compounds alone. However, overall, there is strong evidence that the main hit, UNC10245380A, is a genuine active. First, it has shown well-behaved dose-response curves in 8 independent runs. Second, UNC10245380A has shown selective cell-killing activity in two CIB1-depletion-sensitive cell lines (and no cell-killing activity in a CIB1-depletion-insensitive cell line), which would be unlikely for a fluorescent or quenching compound. Third, many of 24 structural analogs of UNC10245380A have shown varying activity in the TR-FRET assay (a hit expansion result characteristic of true positives).

Once the fragment hotspots were identified, Schrödinger Phase and Glide were used to enumerate virtual hits. This raises the question of how much did FRASE-bot contribute to this. Requesting that a control be done is an onerous request. Furthermore, 3 out of 56 is 5±3%, which is decent for docking —19/56 is 33±5%, which is better than even elaborations of first round hits, albeit with the caveat that not all may be true positives for displacement. So, I sense the answer is that FRASE did contribute. However, if my fellow reviewers have raised similar concerns, please consider this as an underlining of their concerns.

A fully unbiased assessment of a virtual screening workflow (with 0% human intervention) is hardly possible because of the pressure to find hits for the protein of interest while avoiding waste of resource (often on a limited budget). Accurate evaluation of what a specific component of the workflow contributes to the overall success (or the lack thereof) would have been beyond our means. Eventually, to a certain degree, one need to rely on circumstantial evidence. In this study, indeed, the method shows a hit rate that is higher than a typical docking-only protocol. Most recently, FRASE-bot participated in the Critical Assessment of Computational Hit-finding Experiments (CACHE) Challenges #1 and #2, competitions in which 25 teams have been competitively selected by to computationally screen unconventional targets the WDR domain of LRRK2 (#1) and the RNA binding cleft of SARS-CoV-2 NSP13 (#2). Very few of the selected 25 teams succeeded to identify any hits for these two targets. In both Challenges, we found hits that have been confirmed in a panel of orthogonal binding and control assays (the final results for Challenge #1 should be released in mid-January).

Moreover, irrespectively of the comparative performance FRASE-bot+docking vs docking-only, FRASE-bot might prove particularly useful in the context of ultra-large screening collections where the fragments identified by FRASE-bot can be used to pre-screen (by a substructure search) a multi-billion-compound database within hours and use the resulting compound set for pharmacophore search and/or docking (the approach we applied in Challenge #2). Finally, as we point out in the manuscript, unlike docking, FRASE-bot automatically determines putative ligand-binding sites where this task might not be obvious.

Throughout the manuscript "FRASE" is used as a given concept, please correct this for non-linear readers, especially for those who jump into the figure legends.

A brief description and definition of FRASE was added to Introduction, so the reader would not need to search it in the original publication (ref. 29). It reads:

"FRASEs are structural descriptors blending the chemical and protein structure spaces. A single FRASE includes a chemically sound ligand fragment (e.g., a cycle with adjacent acyclic groups) of a protein-bound ligand and the nearby protein residues (within a distance of 4-5 Å). A comprehensive FRASE database was collected from publicly available sources of experimentally determined 3D structures of protein-ligand complexes²⁹."

Please elaborate in the results on what UNC10245204 is, e.g., lab-evolved decapeptide CIB1 binder like UNC10245092 and UNC10245109 or fragment of integrin etc. Also, if it's linear why is the blue Alexa-labelled pearl-string cyclical in the figure.

UNC10245204 is a cyclic peptide -S-Ac-YTTPIWNRFC-NH₂ described in ref. 50 (CIB1 IC₅₀ = 15 nM). As described in Methods, the new tracer ligand for TR-FRET was obtained by merging UNC10245204 with an AlexaFluor 647 dye.

The text mistakenly refers to Figures 2B and 2C as Figure 3. Additionally, the inset in Figure 2C is incorrectly labeled as '3D'.

Corrected

Please specify that the sequence identity to its closest paralogue is AA, it is obvious and correct, but avoids any doubt in skeptical readers.

You probably refer to the text that reads “the closest family member, Calcium and Integrin Binding protein 4 (CIB4), showing only 45% of sequence identity to CIB1”. It is not clear about what needs to be done to avoid any doubt in skeptical readers.

Reviewer #2:

The FRASE-bot protocol described in the manuscript is presented as generally applicable for orphan targets (in contrast with the kinases, where it's relatively easy to de-orphanize a new kinase with structural data of ligands from other kinase families). However, I have some objections regarding the general applicability of this protocol.

1. CIB1 is not truly an orphan target, as there are several known peptides (identified by the same group) targeting this protein. I suspect that the authors have used in the selection process described in this paper some structural information derived from the complexes of these peptides with CIB1. The authors should include in the manuscript a structural comparison of the small molecules inhibitors from this study with the previously identified peptides, with the pharmacophoric features selected during the workflow highlighted.

Claims that CIB1 is an orphan were removed. We added a figure showing the superposed docking pose of UNC10245380 and the co-crystallized structure of the linear peptide UNC10245109 (FWYGAMKALY) (Figure 3C) and a comparative analysis of these structures on page 8 of the manuscript PDF.

Nevertheless, we believe that CIB1 was reasonable proxy of an orphan protein and that the FRASE-bot protocol is a useful tool for finding ligands to genuine orphan target. As mentioned in response to Reviewer 1, we participated in the Critical Assessment of Computational Hit-finding Experiments (CACHE) Challenges #1 and #2, where the Challenge #1 target (WDR domain of LRRK2) was by all accounts an orphan target. Our team applied FRASE-bot and was among a small number who succeeded to identify any hits to those challenging targets (and our hit rate was among the highest). The Challenge #1 results should be released by mid-January.

2. Related to 1., a TR-FRET assay based on these peptides was used for the experimental validation of the initial hits, which would not be possible for a real orphan target.

TR-FRET was a natural first option since it was already available. Otherwise, Differential Scanning Fluorimetry (DSF) and/or Surface Plasmon Resonance (SPR) could have been applied.

3. The FRASE-bot procedure is very complex, involving tools from two different modeling packages (PipelinePilot from BIOVIA and Phase and Glide from Schrödinger). Why it was not possible to use KNIME from Schrödinger instead of PipelinePilot ? Additionally, there are many steps involving human intervention and subjective decisions in the selection process, which makes this protocol difficult to apply to other targets.

The procedure may look complex because we provide a full description of what is under the hood (e.g., it took a long paper to describe the Glide docking algorithm). In principle, the whole workflow applied in this manuscript can be executed with a single push of a button. We do not do this yet because we are still learning how different components of the workflow perform and are improving them, but this will be possible sometime soon.

We developed the initial prototype in Pipeline Pilot because, based on our experience, it is significantly more user-friendly than KNIME and allows saving a lot of time. As specified in Supplemental Information, we share the latest version of FRASE-bot as a Python code making use of public libraries, such as RDkit, Pandas, etc., that can be executed in any interactive environment (e.g., Jupyter Notebook or Colab). However, the Pipeline Pilot protocols, with all intermediary data files, are shared as well to make it possible to reproduce our results.

4. In my opinion, the straightest approach to identify new small molecule inhibitors of CIB1 is to design peptidomimetics, based on the structure of the previously identified peptides. Did the authors attempted this approach ? If yes, what was the outcome ?

For a number of reasons (including lack of funding for a medchem program), we did not try the peptidomimetics approach. Giving a chance to FRASE-bot was an affordable option that eventually benefited both to the method development and hit finding.

Also, it would be useful for the community to specify the names of "collections of ~110K commercially available compounds" (page 8), to avoid their useless evaluation by other groups.

Those are made-to-order libraries purchased by the Center for Integrative Chemical Biology and Drug Discovery (CICBDD) at UNC-Chapel Hill more than 15 years ago. Knowledge of their details is unlikely to be helpful to anyone in the context of this manuscript. However, we added a reference (ref. 73) to the respective CICBDD web page, so that any interested party could find more information about the collections.

5. Machine learning is indeed involved in the first step of the selection process, but its role doesn't justify to use it in the title of the manuscript (docking and pharmacophoric search are also essential for the successful outcome of the selection process).

We modified the title.

p3: "Several 3D structures of apo and ligand bound CIB1" - specify that these are peptidic ligands, not small molecules

This portion of Introduction was modified to address a concern by another reviewer, so this text does not exist anymore.

p4, caption Figure 1A: replace "(vi)" by "(iv)"

Done

Supplementary Excel file:

- replace "Smiles" by "SMILES"

- most SMILES strings are incorrect, please check and replace them with the correct version. For example, for the first structure it should be COC1=CC(=CC=C1)OCCCN2C=C(C3=CC=CC=C32)/C=C\4/C(=O)NC(=O)N(C4=O)CC5=CC=CO5. Also, use Isomeric SMILES instead of Canonical SMILES to avoid losing important stereochemical information.

Done. We generated new SMILES using a different source, RDKit (was Pipeline Pilot).

Reviewer #3:

Using this strategy, the authors identified UNC10245380 as the most promising compound. The authors tested the target hit selectivity of the compound in a set of triple negative breast cancer (TNBC) cells whose CIBP1-dependency for growth were reported by another laboratory. Based on a preliminary cell death data from trypan blue exclusion assay (shown Fig. 3C), and western blot of AKT and ERK phosphorylation (the authors did not show the western blot data), the authors claim to have showed specific cell-killing activity in CIB1-dependent TNBC cells. To substantiate the claim, the authors should perform orthogonal assays for cell death (for example, PARP-cleavage, Annexin V staining) that can be performed with these cell lines.

It would be of great interest to pursue the experimental studies of the lead compound in various functional assays. However, we believe that such an extended investigation would be outside of the scope of this manuscript which is mainly focused on the computational method. Moreover,

The sentence in the Abstract "...showing specific cell-killing activity in CIB1-dependent cancer cells, but not in CIB1-depleted cells" is scientifically not precise. It should be "... CIB1-depletion-insensitive cells" as was used on page 7 of the manuscript.

Corrected

In the paper the current manuscript has referred to (#32, Black et al., Breast Cancer Res Treat, 152:337-346(1915), the authors there did not say that the TNBC cells that are insensitive are depleted of CIBP1 (see western blot in Fig. 1B in that paper where insensitive cells do express CIB1 similar to the sensitive cells). Therefore, it is important to phrase the sentence appropriately.

In the new version of Introduction, the only sentence that refers to ref. 32 is: "The new method was applied to identify potential agents against Triple Negative Breast Cancer (TNBC) by targeting Calcium- and Integrin-Binding Protein 1 (CIB1)". It only suggests that there is some evidence of CIB1 implication in TNBC without making and strong/specific claims. Hopefully, this addresses your concern.

Minor comment: There is no Fig. 3 in the manuscript. Incorrectly, the Fig is labeled Figure 4.

Corrected

REVIEWER COMMENTS

Reviewer #1 (Remarks to the Author):

Excellent. My various concerns have been more than suitably addressed. This is now an useful open tool, which I am glad I got to review.

Re my cryptic _AA I simply meant it was amino acid and not nucleotide sequence identity. Few incorrectly use the latter so I doubt there would be such a diffident reader than needs dissuading.

Reviewer #1 (Remarks on code availability):

All the code is now available.

Licences have been added.

Installation is not in a requirements.txt but is straightforward and in the README.md (which makes sense given tensorflow+cuda installation is driver specific).

I did not run the code, but I skimmed it and it appears to run within a folder without hardcoded absolute paths or similar.

I appreciate that the number of files (10,598) in the compress folder is stated (so I'd unzip it to the node's scratch to avoid degenerating the NFS on a cluster).

Nothing confusing.

Maybe heredoc-style multiline comment in the shell scripts would be nice, but they are documented in the readme.

Reviewer #2 (Remarks to the Author):

In this revised version, the authors have addressed most of my remarks. However, the information added in this version clearly shows that the workflow described is heavily influenced by the prior knowledge of peptide inhibitors UNC10245109 and UNC10245092 and their associated X-ray structures (6OCX and 6OD0).

1. In Figure 3C the authors do not highlight the pharmacophoric features used for screening, as I requested in my previous remarks. Is this because they do not match with the interactions observed ?

2. In the following comments, I will show how the prior knowledge of the peptide inhibitors and their structures have influenced the present work, which could not be successful in the absence of this knowledge. This aspect highlights exactly the limitations of this protocol in addressing orphan targets. Being myself a participant to this kind of challenges, I appreciate the good results obtained by the authors with this protocol for the WDR domain of LRRK2 at the CACHE challenge, but this is not the object of this manuscript.

a. A crystal structure of CIB1 in the apo form (1Y1A) is available, which is the structural information that is typically used for an orphan target (and not a protein-ligand complex structure like 6OCX, which by definition doesn't exist for an orphan target). In the absence of this kind of structure, a homology model may be used, or in these modern days, an AlphaFold2 model (see below).

b. The crystal structures 1Y1A and 6OCX have different relative orientations of the two domains of CIB1, and therefore different dimer interfaces (see the attached Figure R1). Both structures contain calcium ions (colored in green and blue in Figure R1), so the conformational change is not induced by the calcium alone, but very likely by the peptide ligand. However, this is not an information that could be known for a true orphan target.

c. Structure 6OCX (which was used by the authors for the screening described in the present work) is

truncated at the residue 181. The structure 1Y1A (which should have been used in the case of a true orphan target) is complete, with the C-terminal residues 182-191 occupying the same space as the peptide ligand UNC10245109. In other words, the binding site is completely closed in this structure, with no cavity, and the workflow presented in this paper would not be able to identify any molecule targeting this site starting from the structure 1Y1A. The same with the AlphaFold model, which also occludes completely the binding site, with the region 182-191 in a different conformation (Figure R2). Of course, there are techniques allowing the "opening" of the binding site (targeted MD, etc.), but this requires a prior knowledge about the existence of this site which would not be available for a true orphan target.

d. The authors were aware of all these facts (quoting from a previous paper of the same group, <https://doi.org/10.1021/acscchembio.0c00144>: "the 2.1 Å resolution structure revealed that the peptide binds as an α -helix in the H10 pocket, displacing the CIB1 C-terminal H10 helix and causing conformational changes in H7 and H8."), but in this work the structure 6OCX is presented as a convenient choice for simulating an orphan target.

3. The specific selection of pharmacophoric features composing the query from Figure 2C (and not 2D as mentioned in the text) is not justified in any way in the text, very likely this being influenced by the prior knowledge of the binding mode of peptide ligand UNC10245109.

4. In the answer to point 3 in my previous remarks, the authors state that "In principle, the whole workflow applied in this manuscript can be executed with a single push of a button.". How this is compatible with "Visual inspection of the protein structure suggested that centroids forming two groups..." ?

5. The SMILES strings in the supplementary Excel files look correct now, but most of the images containing structures in the first tab are incorrect, showing saturated instead of aromatic rings.

Overall, I consider that the present work was heavily influenced by the prior knowledge of peptide ligands of CIB1 previously reported by the same group and therefore it is not at all representative for studies involving orphan targets, as it is claimed ("we developed FRASE-bot, a computational approach to exploit the full body of 3D structural and SAR data to assemble a ligand in the binding site of any orphan protein").

Reviewer #3 (Remarks to the Author):

The authors have addressed most of the concerns raised in the original review. However, there was no response to the concern regarding the absence of western blot of AKT and ERK phosphorylation. The authors should show the western blot as part of "Supplementary Data" to support their claim "Western blot analysis showed that UNC10245380 inhibited AKT and ERK phosphorylation in CIB1 depletion-sensitive, but not insensitive cell lines".

We thank the reviewers for their feedback that helped us to significantly improve the manuscript. Below is a point-by-point response to the reviewers' concerns (the original reviewers' text is rendered in blue font). In the manuscript file, the new or modified text is highlighted in yellow.

Reviewer #2:

In this revised version, the authors have addressed most of my remarks. However, the information added in this version clearly shows that the workflow described is heavily influenced by the prior knowledge of peptide inhibitors UNC10245109 and UNC10245092 and their associated X-ray structures (6OCX and 6OD0).

Overall, I consider that the present work was heavily influenced by the prior knowledge of peptide ligands of CIB1 previously reported by the same group and therefore it is not at all representative for studies involving orphan targets, as it is claimed ("we developed FRASE-bot, a computational approach to exploit the full body of 3D structural and SAR data to assemble a ligand in the binding site of any orphan protein").

2. In the following comments, I will show how the prior knowledge of the peptide inhibitors and their structures have influenced the present work, which could not be successful in the absence of this knowledge. This aspect highlights exactly the limitations of this protocol in addressing orphan targets. Being myself a participant to this kind of challenges, I appreciate the good results obtained by the authors with this protocol for the WDR domain of LRRK2 at the CACHE challenge, but this is not the object of this manuscript.

We appreciate your thorough, thought-provoking analysis. It made us to adopt a more nuanced view on how generally applicable our approach is. At the same time, in complement to your analysis, the following evidence and rationale need to be considered. First, the CIB1 structure was being studied for more than twenty years and substantial knowledge was accumulated to inform selection of the target structure for small-molecule design even before the latest structures (6OCX and 6OD0) were solved (see details below, in response to more specific structural remarks). Second, there may be a degree of ambiguity in what to call a ligand-orphan target. In the manuscript, we meant a protein with no small-molecule ligands known. Your definition is apparently stricter and means no known ligands of any kind. However, in general, for every potential therapeutic target there is a good deal of information available including structural information about the endogenous interaction partners. There are probably very few pristinely orphan proteins out there (WD40 LRRK2, to which FRASE-bot identified hits in CACHE Challenge, is probably a closest example). To avoid confusion, on page 3, we revised the sentence featuring "... any orphan protein" and provided a specific meaning of "ligand-orphan" in this manuscript. Also, in the Discussion section, on page 9, we added a paragraph on the method's limitations and caveats.

Even though any preexisting structural information (such as, protein-protein or protein-peptide complexes) is always of great help for drug design, it is in no way a promise of a certain small-molecule hit finding. Success still depends on the expertise of drug designers involved and the computation method used.

We agree though that our method is not universally applicable and in case the available structure is some kind of "closed" protein conformation, FRASE screening would fail to identify tractable ligand fragments. That being said, in our experience, more often than not, available structures of ligand-orphan proteins are useful targets for structure-based screening and design.

1. In Figure 3C the authors do not highlight the pharmacophoric features used for screening, as I requested in my previous remarks. Is this because they do not match with the interactions observed?

Done. No, that was not the reason. For UNC10245380, three of four matching features seem to contribute significantly to its affinity and one (HBA on the left side of the figure) probably does not, though we would only know for sure after a detailed SAR exploration (which is beyond the scope of this manuscript).

a. A crystal structure of CIB1 in the apo form (1Y1A) is available, which is the structural information that is typically used for an orphan target (and not a protein-ligand complex structure like 6OCX, which by definition doesn't exist for an orphan target). In the absence of this kind of structure, a homology model may be used, or in these modern days, an AlphaFold2 model (see below).

b. The crystal structures 1Y1A and 6OCX have different relative orientations of the two domains of CIB1, and therefore different dimer interfaces (see the attached Figure R1). Both structures contain calcium ions (colored in green and blue

in Figure R1), so the conformational change is not induced by the calcium alone, but very likely by the peptide ligand. However, this is not an information that could be known for a true orphan target.

c. Structure 6OCX (which was used by the authors for the screening described in the present work) is truncated at the residue 181. The structure 1Y1A (which should have been used in the case of a true orphan target) is complete, with the C-terminal residues 182-191 occupying the same space as the peptide ligand UNC10245109. In other words, the binding site is completely closed in this structure, with no cavity, and the workflow presented in this paper would not be able to identify any molecule targeting this site starting from the structure 1Y1A. The same with the AlphaFold model, which also occludes completely the binding site, with the region 182-191 in a different conformation (Figure R2). Of course, there are techniques allowing the "opening" of the binding site (targeted MD, etc.), but this requires a prior knowledge about the existence of this site which would not be available for a true orphan target.

d. The authors were aware of all these facts (quoting from a previous paper of the same group, <https://doi.org/10.1021/acscchembio.0c00144>: "the 2.1 Å resolution structure revealed that the peptide binds as an α -helix in the H10 pocket, displacing the CIB1 C-terminal H10 helix and causing conformational changes in H7 and H8."), but in this work the structure 6OCX is presented as a convenient choice for simulating an orphan target.

Before the structures 6OCX and 6OD0 were solved, a large body of evidence and knowledge has been generated on the CIB1 structure and function. In particular, peptide-bound complex was homology-modelled using integrin-binding proteins, such as calcineurin, in complex with the Integrin α IIb cytoplasmic domain as a structural template. One such structure, 1DGU, deposited to PDB [1] shows a fold strikingly similar to those of 6OCX/6OD0 (Fig. 1A). This structural hypothesis of CIB- α IIb interactions was further corroborated by mutagenesis [2]. These and other studies led to a hypothesis of "open" (i.e., integrin-binding) and "closed" (e.g., that in 1Y1A) conformations [3]. Later, a solution NMR structure was solved showing an apparently open conformation similar to those in 6OCX/6OD0 (Fig. 1B) [4]. There was a consensus that the CIB1 C-terminal residues (as seen in apo structures) occlude the binding pocket and must be either removed or moved aside (as in 1DGU) when modeling peptide-bound structures (e.g., [5]). Eventually, the experimentally solved peptide-bound structures 6OCX and 6OD0 were not a revelation, but rather a useful corroboration of the previously accumulated knowledge. Given all the above, even in the absence of 6OCX/6OD0, 1Y1A would have been the last structure to consider for our screening project. We would have rather made use of 2LM5 or 1DGU (or an MD structure equilibrated in complex with α IIb peptide).

Figure 1. A. 6OD0 (magenta) and 1DGU (brown); B. 6OD0 (magenta) and 1DGU (violet)

[1] P.M. Hwang, H.J. Vogel, Structures of the platelet calcium- and integrin-binding protein and the α IIb-integrin cytoplasmic domain suggest a mechanism for calcium-regulated recognition; homology modelling and NMR studies. (2000) *J Mol Recognit*, 13: 83-92

[2], W.T. Barry et al., Molecular Basis of CIB Binding to the Integrin α IIb Cytoplasmic Domain, *J Biol Chem*, 277, 32, 28877 – 28883

[3] H.R. Gentry et al., Structural and Biochemical Characterization of CIB1 Delineates a New Family of EF-hand-containing Proteins, *J Biol Chem*, 280, 9, 8407 – 8415

[4] H. Huang and H.J. Vogel, Structural Basis for the Activation of Platelet Integrin α IIb β 3 by Calcium- and Integrin-Binding Protein 1, *J Amer Chem Soc*, 2012 134 (8), 3864-3872

[5] T.C. Freeman, J.L. Black, H.G. Bray, O. Dagliyan, Y.I. Wu, A. Tripathy, N. V. Dokholyan, T.M. Leisner, and L.V. Parise, Identification of Novel Integrin Binding Partners for Calcium and Integrin Binding Protein 1 (CIB1): Structural and Thermodynamic Basis of CIB1 Promiscuity, *Biochemistry*, 2013 52 (40), 7082-7090

3. The specific selection of pharmacophoric features composing the query from Figure 2C (and not 2D as mentioned in the text) is not justified in any way in the text, very likely this being influenced by the prior knowledge of the binding mode of peptide ligand UNC10245109.

We added new text on pages 6/7 (at the end of the section "FRASE database screening and converting hit fragments ...") to explain the rationale of choosing centroids 1, 2, 6, and 8 as features for the pharmacophore query. Also, keep in mind that centroids 1, 2, 3, 5, 6, and 8 are located in a deep cavity (a strong reason on its own) that for the last 20 years was considered as a putative binding pocket for the integrin α IIb domain. That is, we did not need the 6OCX structure to have a certain preference for this pocket. Then again, one may argue that due to that prior knowledge CIB1 cannot be called a ligand-orphan target, but this would be a matter of semantics.

4. In the answer to point 3 in my previous remarks, the authors state that "In principle, the whole workflow applied in this manuscript can be executed with a single push of a button.". How this is compatible with "Visual inspection of the protein structure suggested that centroids forming two groups..."?

We practice visual inspection while doing any kind of virtual screening. But those who neglect this step in other types of virtual screening may also skip it in FRASE-based screening and rely on formal quantitative criteria, such as cluster size or the degree of buried'ness when picking features for the final pharmacophore.

5. The SMILES strings in the supplementary Excel files look correct now, but most of the images containing structures in the first tab are incorrect, showing saturated instead of aromatic rings.

The structures were fixed, and a new version of the spreadsheet uploaded.

Reviewer #3:

The authors have addressed most of the concerns raised in the original review. However, there was no response to the concern regarding the absence of western blot of AKT and ERK phosphorylation. The authors should show the western blot as part of "Supplementary Data" to support their claim "Western blot analysis showed that UNC10245380 inhibited AKT and ERK phosphorylation in CIB1 depletion-sensitive, but not insensitive cell lines".

The Western blot image was added as a Supplementary Figure 1.

REVIEWERS' COMMENTS

Reviewer #2 (Remarks to the Author):

The authors have provided detailed answers to my remarks. I agree with most of their reasoning, which highlights however even more the previous knowledge used in the protocol described in this manuscript. I think that FRASE-bot is a useful tool to be incorporated in a more general workflow for the design of new active compounds, but the example presented in this manuscript (with CIB1 as target) doesn't evidence enough its unique capabilities compared with standard approaches in drug design. Probably presenting the use of this tool with a true orphan target, e.g. the WD40 repeat (WDR) domain of LRRK2 from the CACHE Challenge, would have illustrated better its powerfulness.

I realize that we will not reach a consensus regarding the definition of an orphan target. In my view, there is not such a concept as "ligand-orphan protein" with known synthetic small peptide inhibitors. Therefore, I still consider that CIB1 is not an orphan target.

We thank the reviewer for their feedback that helped us to significantly improve the manuscript. Below is a point-by-point response to the reviewer's concerns (the original reviewers' text is rendered in blue font).

Reviewer #2:

The authors have provided detailed answers to my remarks. I agree with most of their reasoning, which highlights however even more the previous knowledge used in the protocol described in this manuscript. I think that FRASE-bot is a useful tool to be incorporated in a more general workflow for the design of new active compounds, but the example presented in this manuscript (with CIB1 as target) doesn't evidence enough its unique capabilities compared with standard approaches in drug design. Probably presenting the use of this tool with a true orphan target, e.g. the WD40 repeat (WDR) domain of LRRK2 from the CACHE Challenge, would have illustrated better its powerfulness.

Participation in CACHE Challenge #1 is described in the Discussion section.

I realize that we will not reach a consensus regarding the definition of an orphan target. In my view, there is not such a concept as "ligand-orphan protein" with known synthetic small peptide inhibitors. Therefore, I still consider that CIB1 is not an orphan target.

All occurrences of the term "orphan" were removed from the manuscript.